# Interpreting Answers to Yes-No Questions in User-Generated Content

**Shivam Mathur,**[1*] **Keun Hee Park,**[2] **Dhivya Chinnappa,**[3]
**Saketh Kotamraju,**[4] and **Eduardo Blanco**[5]

[1]Walmart Global Tech, [2]Arizona State University, [3]JP Morgan Chase & Co.,
[4]University of Texas at Austin, [5]University of Arizona
`shivam.mathur@walmart.com, kpark53@asu.edu,`
`dhivya.infant@gmail.com, saketh.k@utexas.edu,`
`eduardoblanco@arizona.edu`

## Abstract

Interpreting answers to yes-no questions in social media is difficult. *Yes* and *no* keywords are uncommon, and the few answers that include them are rarely to be interpreted what the keywords suggest. In this paper, we present a new corpus of 4,442 yes-no question-answer pairs from Twitter. We discuss linguistic characteristics of answers whose interpretation is *yes* or *no*, as well as answers whose interpretation is *unknown*. We show that large language models are far from solving this problem, even after fine-tuning and blending other corpora for the same problem but outside social media.

## 1 Introduction

Social media has become a global town hall where people discuss whatever they care about in real time. Despite its challenges and potential misuse, the language of social media can be used to approach many problems, including mental health (Harrigian et al., 2020), rumor detection (Ma and Gao, 2020), and fake news (Mehta et al., 2022).

There is extensive literature on question answering, but few previous efforts work with social media. In particular, yes-no questions are unexplored in social media. Figuring out the correct interpretation of answers to yes-no questions (*yes*, *no*, or somewhere in the middle) from formal texts (Clark et al., 2019), short texts (Louis et al., 2020), and conversations (Choi et al., 2018; Reddy et al., 2019) is challenging. This is the case in any domain, but especially in social media, where informal language and irony are common. Additionally, unlike previous work on yes-no questions outside social media, we find that *yes* and *no* keywords are not only rare in answers, but also that they are poor indicators of the correct interpretation of the answer.

Figure 1: Yes-No question from Twitter and three answers. Interpreting answers cannot be reduced to finding keywords (e.g., *yes*, *of course*). Answers ought to be interpret as *yes* (despite the negation), *no* (commonsense and reasoning are needed), and *unknown* respectively.

Consider the Twitter thread in Figure 1. The question is mundane: whether people preheat the oven before using it. The answers, however, are complicated to decipher. The first answer states that it is *very strange* that somebody else (the author's girlfriend) does not preheat the oven. Note that the correct interpretations is *yes* (or *probably yes*) despite the negation. The second answer explains why it is unnecessary to preheat the oven—without explicitly saying so—thus the correct interpretation is *no*. The third answer includes both *yes* and *no* yet the correct interpretation is *unknown*, as the author does not commit to either option.

In this paper, we tackle the problem of interpreting *natural* answers to yes-no questions posted on Twitter by real users, as exemplified above. The main contributions are:[1] (a) Twitter-YN, a new corpus of 4,442 yes-no questions and answers from Twitter along with their interpretations; (b) analysis discussing the language used in the answers, and in particular, showing that keyword presence in answers is a bad predictor of their interpretations, (c) experimental results showing that the problem is challenging—even for large language models, including GPT3—and blending with related corpora is useful; and (d) qualitative analysis of the most common errors made by our best model.

---

*Work done while at Arizona State University.

[1]Twitter-YN and implementation available at https://github.com/shivammathur33/twitter-yn.

**Motivation** Traditionally, question answering assumes that there are (a) correct answers to questions or (b) no answers (Kwiatkowski et al., 2019). In the work presented in this paper, answers are available—the problem is to interpret what they mean. As Figure 1 shows, answers capture personal preferences rather than correct answers supported by commonsense or common practice (e.g., oven manuals state that preheating is recommended). Interpreting answers to yes-no questions opens the door to multiple applications. For example, dialogue systems for social media (Sheng et al., 2021) must understand answers to yes-no question to avoid inconsistencies (e.g., praising the author of the second reply in Figure 1 for preheating the oven is nonsensical). Aggregating interpretations could also assist in user polling (Lampos et al., 2013). As the popularity of chat-based customer support grows (Cui et al., 2017), automatic interpretation of answers to yes-no questions could enhance the customer experience. Lastly, interactive question answering and search, in which systems ask clarification questions to better address users' information needs, would benefit from this work (Li et al., 2017). Clarification questions are often yes-no questions, and as we shall see, people rarely use *yes*, *no* or similar keywords in their answers. Rather, people answer with intricate justifications that are challenging to interpret.

## 2 Previous Work

**Question Answering Outside Social Media** has a long tradition. There are several corpora, including some that require commonsense (Talmor et al., 2019) or understanding tables (Cheng et al., 2022), simulate open-book exams (Mihaylov et al., 2018), and include questions submitted by "real" users (Yang et al., 2015; Kwiatkowski et al., 2019). Models include those specialized to infer answers from large documents (Liu et al., 2020), data augmentation (Khashabi et al., 2020), multilingual models (Yasunaga et al., 2021), and efforts to build multilingual models (Lewis et al., 2020). None of them target social media or yes-no questions.

**Yes-No Questions** and interpreting their answers have been studied for decades. Several works target exclusive answers not containing *yes* or *no* (Green and Carberry, 1999; Hockey et al., 1997). Rossen-Knill et al. (1997) work with yes-no questions in the context of navigating maps (Carletta et al., 1997) and find that answers correlate with actions

taken. More recent works (de Marneffe et al., 2009) target questions from SWDA (Stolcke et al., 2000). Clark et al. (2019) present yes-no questions submitted to a search engine, and Louis et al. (2020) present a corpus of crowdsourced yes-no questions and answers. Several efforts work with yes-no questions in dialogues, a domain in which they are common. Annotation efforts include crowdsourced dialogues (Reddy et al., 2019; Choi et al., 2018) and transcripts of phone conversations (Sanagavarapu et al., 2022) and *Friends* (Damgaard et al., 2021).

**Question Answering in the Social Media Domain** is mostly unexplored. TweetQA is the only corpus available (Xiong et al., 2019). It consists of over 13,000 questions and answers about a tweet. Both the questions and answers were written by crowdworkers, and the creation process ensured that it does *not include any yes-no question*. In this paper, we target yes-no questions in social media for the first time. We do not tackle the problem of extracting answers to questions. Rather, we tackle the problem of *interpreting* answers to yes-no questions, where both questions and answers are posted by genuine social media users rather than paid crowdworkers.

## 3 Twitter-YN: A Collection of Yes-No Questions and Answers from Twitter

Since previous research has not targeted yes-no questions in social media, our first step it to create a new corpus, Twitter-YN. Retrieving yes-no questions and their answers takes into account the intricacies of Twitter, while the annotation guidelines are adapted from previous work.

### 3.1 Retrieving Yes-No Questions and Answers

We define a battery of rules to identify yes-no questions. These rules were defined after exploring existing corpora with yes-no questions (Section 2), but we adapted them to Twitter. Our desideratum is to collect a variety of yes-no questions, so we prioritize precision over recall. This way, the annotation process is centered around interpreting answers rather than selecting yes-no questions from the Twitter fire hose. The process is as follows:

1. Select tweets that (a) contain a question mark ('?') and the bigram *do you*, *is it*, *does it* or *can it*; and (b) do not contain wh-words (*where*, *who*, *why*, *how*, *whom*, *which*, *when*, and *whose*). Let us name the text between the bigram and '?' a *candidate* yes-no question.

| Question | Answer | Intpn. |
|---|---|---|
| Do you think it is rude to say to somebody you would look better if you lost weight? | Not only rude, but potentially lethal depending on who you say it to! | *yes* |
| Is it ok to read a book at a bar? | Sure, if you don't mind getting beaten up. | *no* |
| Is it big coat weather? | If the news is reporting weather then it's got to be cold 🌨️ 🏙️ | *prob. yes* |
| Is it bad if I brake with my left foot while driving? | It's how racecar drivers do it 🤷 | *prob. no* |
| Do you want Daylight savings time permanent? | Don't care we have bigger issues to worry about | *unknown* |

Table 1: Examples of yes-no questions and answers from our corpus. Some answers include negative keywords but their interpretation is *yes* (first example), whereas others include positive keywords but their interpretation is *no* (second example). Answers imposing conditions are interpreted *probably yes* or *probably no* (third and fourth examples). Negation does not necessarily indicate that the author leans to either *yes* or *no* (fifth example).

2. Exclude candidate questions unless they (a) are between 3 and 100 tokens long; (b) do not span more than one paragraph; and (c) do not contain named entities or numbers.
3. Exclude candidate questions with links, hashtags, or from unverified accounts.

The first step identifies likely yes-no questions. We found that avoiding other questions with wh-words is critical to increase precision. The second step disregards short and long questions. The reason to discard questions with named entities is to minimize subjective opinions and biases during the annotation process—most yes-no questions involving named entities ask for opinions about the entity (usually a celebrity). We discard questions with numbers because the majority are about cryptocurrencies or stocks and we want to avoid having tweets about the financial domain (out of a random sample of 100 questions, 68% of questions with numbers were about cryptocurrencies or stocks). The third step increases precision by considering only verified accounts, which are rarely spam. Avoiding hashtags and links allows us to avoid answers whose interpretation may require external information including trending topics.

Once yes-no questions are identified, we consider as answers all replies except those that contain links, more than one user mention (@user), no verb, less than 6 or more than 30 tokens, or question marks ('?'). The rationale behind these filters is to avoid answers whose interpretation require external information (links) or are unresponsive. We do not discard answers with *yes* or *no* keywords because as we shall see (and perhaps unintuitively) they are not straightforward to interpret (Section 4).

**Temporal Spans: Old and New** We collect questions and their answers from two temporal spans: older (years 2014–2021) and newer (first half of 2022). Our rationale is to conduct experiments in a realistic setting, i.e., training and testing with questions posted in non-overlapping temporal spans.

The retrieval process resulted in many yes-no question-answer pairs. We set to annotate 4,500 pairs. Because we wanted to annotate all answers to the selected questions across both temporal spans, Twitter-YN consists of 4,442 yes-no question-answer pairs (old: 2,200, new: 2,242). The total number of questions are 1,200 (3.7 answers per question on average).

### 3.2 Interpreting Answers to Yes-No Questions

After collecting yes-no questions and answers, we manually annotate their interpretations. We work with the five interpretations exemplified in Table 1. Our definitions are adapted from previous work (de Marneffe et al., 2009; Louis et al., 2020; Sanagavarapu et al., 2022) and summarized as follows:

- *Yes*: The answer ought to be interpreted as *yes without reservations*. In the example, the answer doubles down on the comment that it is rude to suggest that somebody would look better if they lost weight (despite also using *not*) by characterizing the comment as lethal.
- *No*: The answer ought to be interpreted as *no without reservations*. In the example, the author uses irony to effectively state that one should not read books in bars.
- *Probably yes*: The answer is *yes* under certain condition(s) (including time constraints) or it shows arguments for *yes* from a third party. In the example, the answer relies on the weather report to mean that it will be big coat weather.

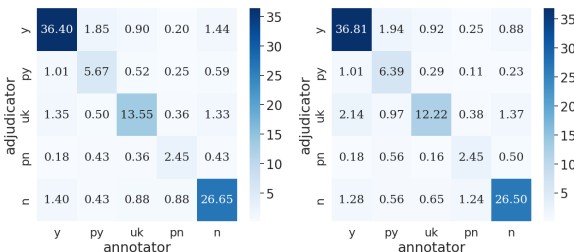

Figure 2: Differences between adjudicated interpretations and interpretations by each annotator. There are few disagreements; the diagonals (i.e., the adjudicated interpretation and the one by each annotator) account for 84.72% and 84.37%. Additionally, most disagreements are minor: they are between (a) *yes* (*no*) and *probably yes* (*probably no*) or (b) *unknown* and another label.

- *Probably no*: The answer is a *no* under certain condition(s) or shows arguments for *no* held by a third party, or the author shows hesitancy. In the example, the author indicates that it is *not bad* to brake with your left foot while driving if you are a racecar driver.
- *Unknown*: The answer disregards the question (e.g., changes topics) or addresses the question without an inclination toward *yes* or *no*. In the example, the author states that the question is irrelevant without answering it.

Appendix A presents the detailed guidelines.

**Annotation Process and Quality** We recruited four graduate students to participate in the definition of the guidelines and conduct the annotations. The 4,442 yes-no question-answers pairs were divided into batches of 100. Each batch was annotated by two annotators, and disagreements were resolved by an adjudicator. In order to ensure quality, we calculated inter-annotator agreement using linearly weighted Cohen's $\kappa$. Overall agreement was $\kappa = 0.68$. Note that $\kappa$ coefficients above 0.6 are considered *substantial* (Artstein and Poesio, 2008) (above 0.8 are considered *nearly* perfect). Figure 2 shows the disagreements between the interpretations by annotators and the adjudicator. Few disagreements raise concerns. The percentage of disagreements between *yes* and *no* is small: 1.40, 1.44, 1.28, and 0.88. We refer the reader to the Data Statement in Appendix B for further details about Twitter-YN.

## 4 Analyzing Twitter-YN

Twitter-YN is an unbalanced dataset: the interpretation of 70% of answers is either *yes* or *no* (Table 2). More interestingly, few answers contain a *yes* or *no*

|  |  | % Contains Keyword | |
| Interpretation | % | positive | negative |
| --- | --- | --- | --- |
| *yes* | 40.79 | 13.42 | 11.98 |
| *no* | 30.23 | 2.23 | 18.91 |
| *probably yes* | 8.04 | 1.33 | 2.43 |
| *probably no* | 3.85 | 0.33 | 2.39 |
| *unknown* | 17.09 | 0.92 | 5.70 |

Table 2: Distribution of interpretations in our corpus (Columns 1 and 2) and keyword-based analysis (Column 3 and 4). Only 13.42% (18.91%) of answers with *yes* (*no*) interpretations include a positive (negative) keyword. Interpreting answers cannot be reduced to checking for positive or negative keywords.

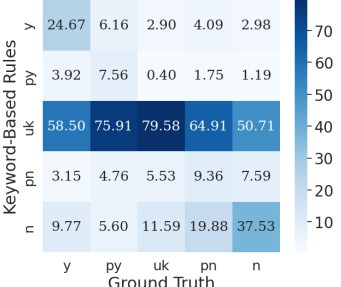

Figure 3: Heatmap comparing the interpretations predicted with keyword-based rules (Section 4) and the ground truth. Rules are insufficient for this problem.

keyword, and those that do often are not interpreted what the keyword suggests. For example, 11.98% of answers labeled *yes* contain a negative keyword, while only 13.42% contain a positive keyword.

**Are Keywords Enough to Interpret Answers?** No, they are not. We tried simple keyword-based rules to attempt to predict the interpretations of answers. The rules we consider first check for *yes* (*yes*, *yea*, *yeah*, *sure*, *right*, *you bet*, *of course*, *certainly*, *definitely*, or *uh uh*), *no* (*no*, *nope*, *no way*, *never*, or *n't*), and *uncertainty* (*maybe*, *may*, *sometimes*, *can*, *perhaps*, or *if*) keywords. Then they return (a) *yes* (or *no*) if we only found keywords for *yes* (or *no*), (b) *probably yes* (or *probably no*) if we found keywords for *uncertainty* and *yes* (or *no*), (c) *unknown* otherwise. As Figure 3 shows, such a strategy does not work. Many instances are wrongly predicted *unknown* because no keyword is present, and keywords also fail to distinguish between *no* and *probably no* interpretations.

### 4.1 Linguistic Analysis

Despite all yes-no questions include one of four bigrams (*do you*, *is it*, *does it* or *can it*, Section 3.1),

| Bigram | Verb (base form, top-10) |
|---|---|
| *do you* (66.4%) | *think* (9.5), *have* (7.8), *believe* (5.4), *want* (3.7), *use* (2.6), *get* (2.4), *do* (2.3), *like* (2.1), *go* (1.8), and *feel* (1.7) |
| *is it* (30.6%) | *have* (5.3), *go* (4.9), *drive* (3.6), *ask* (3.3), *look* (3.0), *put* (2.5), *want* (2.3), *use* (2.2), *wear* (2.0), and *eat* (1.8) |
| *does it* (2.5%) | *make* (19.1), *have* (6.0), *hear* (5.5), *matter* (5.5), *get* (5.1), *take* (4.7), *tear* (4.7), *count* (4.3), *work* (4.3), and *exist* (3.0) |
| *can it* (0.5%) | *offer* (32.0), *ask* (32.0), *understand* (32.0), *reply* (2.0), and *wait* (2.0) |

Table 3: Most common bigrams and verbs in the questions. Most questions include the bigrams *do you* or *is it* (97%). The verb percentages show that the questions are diverse. Indeed, the top-10 most frequent verbs with *do you* only account for 39.3% of questions with *do you*.

and 97% include *do you* or *is it*, Twitter-YN has a wide variety of questions. Indeed, the distribution of verbs has a long tail (Table 3). For example, less than 10% of questions containing *do you* have the verb *think*, Further, the the top-10 most frequent verbs account for 39.3% of questions starting with *do you*. Thus, the questions in Twitter-YN are diverse from a lexical point of view.

We also conduct an analysis to shed light on the language used in answers to yes-no questions (Table 4). Verbs and pronouns are identified with spaCy (Honnibal and Montani, 2017). We use WordNet lexical files (Miller, 1995) as verb classes, the negation cue detector by Hossain et al. (2020), and the sentiment lexicon by Crossley et al. (2017).

First, we compare answers interpreted as *yes* and *no*. We observe that cardinal numbers are more common in *yes* rather than *no* interpretations. Object pronoun *me* is also more common with *yes* interpretations; most of the time it is used to describe personal experiences. Despite the low performance of the keyword-based rules, negation cues and negative sentiment are more frequent with *no* interpretations. Finally, emojis are more frequent in humorous answers meaning *yes* rather than *no*.

Second, we compare *unknown* and other interpretations (i.e., when the author leans towards neither *yes* nor *no*). We observe that longer answers are less likely to be *unknown*, while more verbs indicate *unknown*. All verb classes are indicators of *unknown*. For example, communication (say, tell, etc.), creation (bake, perform etc.), motion (walk,

| Linguistic Feature | *yes* vs. *no* | *unk* vs other |
|---|---|---|
| Tokens | | ↓ |
| Cardinal numbers | ↑ | |
| Verbs | | ↑↑↑ |
| verb.communication | | ↑↑↑ |
| verb.creation | | ↑ |
| verb.motion | | ↑ |
| verb.perception | | ↑↑ |
| *would* | ↓↓ | |
| *must* | | ↑ |
| *shall* | | ↑↑ |
| 1st person pronoun | | ↓↓ |
| *I* | | ↓↓↓ |
| *me* | ↑↑ | |
| 2nd person pronoun | | ↑↑↑ |
| *he* | | ↑ |
| *you* | | ↑↑↑ |
| Negation cues | ↓↓↓ | ↓↓↓ |
| Negative sentiment | ↓↓↓ | |
| Emojis | ↑↑↑ | |

Table 4: Linguistic analysis comparing answers interpreted as (a) *yes* and *no* and (b) *unknown* and the other labels. Number of arrows indicate the p-value (t-test; one: $p<0.05$, two: $p<0.01$, and three: $p<0.001$). Arrow direction indicates whether higher counts correlate with the first interpretation (*yes* or *unknown*) or the second.

fly etc.), and perception (see, hear etc.) verbs appear more often with *unknown* answers. Finally, *unknown* answers rarely have negation cues.

## 5 Automatically Interpretation of Answers to Yes-No Questions

We experiment with simple baselines, RoBERTa-based classifiers, and GPT3 (zero- and few-shot).

**Experimental settings** Randomly splitting Twitter-YN into train, validation, and test splits would lead to valid but unrealistic results. Indeed, in the real world a model to interpret answers to yes-no questions ought to be trained with (a) answers to *different questions* than the ones used for evaluation, and (b) answers to questions from *non-overlapping temporal spans*.

In the main paper, we experiment with the most realistic setting: *unmatched question and unmatched temporal span* (train: 70%, validation: 15%, test: 15%). We refer the reader to Appendix C for the other two settings: *matched question and matched temporal span* and *unmatched question and matched temporal span*.

**Baselines** We present three baselines. The *majority class* always chooses the most frequent label in the training split (i.e., *yes*). The *keyword-based rules* are the rules presented in Section 4. We also experiment with a version of the rules in which the lexicon of negation cues is replaced with a *negation cue detector* (Hossain et al., 2020).

## 5.1 RoBERTa-based Classifiers

We experiment with RoBERTa (Liu et al., 2019) as released by TensorFlow (Abadi et al., 2015) Hub. Hyperparameters were tuned using the train and validation splits, and we report results with the test split. We refer readers to Appendix D for details about hyperparameters and the tuning process.

**Pretraining and Blending** In addition to training with our corpus, Twitter-YN, we also explore combining related corpora. The most straightforward approach is pretraining with other corpora and then fine-tuning with Twitter-YN. Doing so is sound but obtains worse results (Appendix E) than blending, so we will focus here on the latter.

Pretraining could be seen as a two-step fine-tuning process: first with related corpora and then with Twitter-YN. On the other hand, blending (Shnarch et al., 2018) combines both during the same training step. Blending starts fine-tuning with the combination of both and decreases the portion of instances from related corpora after each epoch by an $\alpha$ ratio. In other words, blending starts tuning with many yes-no questions (from the related corpus and Twitter-YN) and finishes only with the ones in Twitter-YN. The $\alpha$ hyperparameter is tuned like any other hyperparameter; see Appendix D.

We experiment with pretraining and blending with the following corpora. All of them include yes-no questions and manual annotations indicating the interpretation of their answers. However, none of them are in the Twitter domain:

- BOOLQ (Clark et al., 2019), 16,000 natural yes-no questions submitted to a search engine and (potential) answers from Wikipedia;
- Circa (Louis et al., 2020), 34,000 yes-no questions and answers falling into 10 predefined scenarios and written by crowdworkers;
- SWDA-IA (Sanagavarapu et al., 2022), ≈2,500 yes-no questions from transcriptions of telephone conversations; and
- FRIENDS-QIA (Damgaard et al., 2021), ≈6,000 yes-no questions from scripts of the popular Friends TV show.

**Prompt-Derived Knowledge.** Liu et al., 2022 have shown that integrating knowledge derived from GPT3 in question-answering models is beneficial. We follow a similar approach by prompting GPT3 to generate a *disambiguation text* given a question-answer pair from Twitter-YN. Then we complement the input to the RoBERTa-based classifier with the disambiguation text. For example, given Q: *Do you still trust anything the media says?* A: *Modern media can make a collective lie to become the truth*, GPT3 generates the following disambiguation text: *Mentions how media can manipulate the truth, they do not trust anything the media says*. We refer the reader to Appendix G for the specific prompt and examples of correct and nonsensical disambiguation texts.

## 5.2 GPT3: Zero- and Few-Shot

Given the success of of large language models and prompt engineering in many tasks (Brown et al., 2020), we are keen to see whether they are successful at interpreting answers to yes-no questions from Twitter. We experiment with various prompt-based approaches and GPT3. Our prompts do not exactly follow previous work, but they are inspired by previous work (Liu et al., 2021):

**Zero-Shot.** We use prefix prompts (Li and Liang, 2021; Lester et al., 2021) since such prompts work well with generative language models such as GPT3 (Liu et al., 2021). As shown by Efrat and Levy, 2020, we also experiment with the same zero-shot prompt including the annotation guidelines.

**Few-Shot.** We experiment with longer versions of the zero-shot prompts that include two examples from the training split per label (10 examples total; two versions: with and without the guidelines).

We refer the reader to Appendix G for (a) additional details about the prompts and examples and (b) specifics about the GPT3 experimental setup.

## 6 Results and Analysis

We present results in the *unmatched questions and unmatched temporal span* setting in Table 5. Appendix C presents the results in the other two settings. We will focus the discussion on average F1 scores as Twitter-YN is unbalanced (Section 4).

Regarding the baselines, using a negation cue detector to identify *no* keywords rather than our predefined list is detrimental (F1: 0.27 vs. 0.37), especially with *yes* label. This result leads to the conclusion that many answers with ground truth

| | All | | | | yes | no | pyes | pno | unk |
|---|---|---|---|---|---|---|---|---|---|
| | Accuracy | P | R | F1 | F1 | F1 | F1 | F1 | F1 |
| Baseline, Majority Class | 0.39 | 0.15 | 0.39 | 0.22 | 0.56 | 0.00 | 0.00 | 0.00 | 0.00 |
| Baseline, Keyword-Based Rules | 0.37 | 0.51 | 0.37 | 0.37 | 0.40 | 0.48 | 0.13 | 0.15 | 0.34 |
|    with negation cue detector | 0.30 | 0.48 | 0.30 | 0.27 | 0.22 | 0.44 | 0.08 | 0.18 | 0.33 |
| RoBERTa trained with yes-no questions corpora | | | | | | | | | |
|   BoolQ | 0.43 | 0.29 | 0.43 | 0.32 | 0.58 | 0.35 | 0.00 | 0.00 | 0.00 |
|   Circa | 0.56 | 0.56 | 0.56 | 0.53 | 0.70 | 0.59 | 0.32 | 0.06 | 0.35 |
|   SwDA-IA | 0.50 | 0.53 | 0.50 | 0.47 | 0.67 | 0.52 | 0.26 | 0.02 | 0.15 |
|   Friends-QIA | 0.53 | 0.53 | 0.53 | 0.51 | 0.67 | 0.55 | 0.22 | 0.00 | 0.44 |
|   Twitter-YN (our corpus) | | | | | | | | | |
|     Question only | 0.38 | 0.41 | 0.38 | 0.29 | 0.53 | 0.28 | 0.00 | 0.00 | 0.02 |
|     Answer only | 0.53 | 0.50 | 0.53 | 0.48 | 0.65 | 0.56 | 0.19 | 0.00 | 0.32 |
|     Question and Answer | 0.58 | 0.58 | 0.58 | 0.56 | 0.68 | 0.61 | 0.39 | 0.22 | 0.40 |
| RoBERTa blending Twitter-YN | | | | | | | | | |
|   BoolQ ($\alpha = 0.5$) | 0.57 | 0.45 | 0.57 | 0.55 | 0.70 | 0.60 | 0.31 | 0.06 | 0.44 |
|   Circa ($\alpha = 0.5$) | 0.60 | 0.59 | 0.60 | **0.58**† | 0.72 | 0.65 | 0.37 | 0.25 | 0.44 |
|     +disambiguation text (GPT3) | 0.62 | 0.62 | 0.62 | **0.61**∗† | 0.75 | 0.67 | 0.35 | 0.22 | 0.57 |
|   SWDA-IA ($\alpha = 0.5$) | 0.56 | 0.57 | 0.56 | 0.55 | 0.67 | 0.59 | 0.36 | 0.22 | 0.44 |
|   Friends-QIA ($\alpha = 0.8$) | 0.60 | 0.59 | 0.60 | **0.58**† | 0.71 | 0.63 | 0.38 | 0.20 | 0.45 |
|     +disambiguation text (GPT3) | 0.56 | 0.56 | 0.56 | 0.56 | 0.70 | 0.63 | 0.36 | 0.24 | 0.37 |
| GPT3, Zero-Shot | 0.44 | 0.55 | 0.44 | 0.46 | 0.56 | 0.51 | 0.16 | 0.13 | 0.43 |
| GPT3, Few-Shot | 0.54 | 0.55 | 0.54 | 0.53 | 0.67 | 0.55 | 0.14 | 0.30 | 0.49 |

Table 5: Results in the *unmatched question and unmatched temporal span* setting. Only training with other corpora obtains modest results, and taking into account the question in addition to the answer is beneficial (Twitter-YN, F1: 0.56 vs. 0.48). Blending Circa and Friends-QIA is beneficial (F1: 0.58 vs. 0.56). Adding the disambiguation text from GPT3 with Circa is also beneficial (F1: 0.61). Few-Shot prompt with GPT3 obtains results close to a supervised model trained with Twitter-YN (F1: 0.53 vs. 0.56), but underperforms blending Circa using the disambiguation text from GPT3 (0.61). Statistical significance with respect to training with Twitter-YN (Q and A) is indicated with '∗' and with respect to few-shot GPT3 with '†' (McNemar's Test (McNemar, 1947), p<0.05).

interpretation *yes* include negation cues such as affixes *im-* and *-less*, which are identified by the cue detector but are not in our list. Zero- and Few-Shot GPT3 outperforms the baselines, and obtains moderately lower results than the supervised model trained on Twitter-YN (Question and Answer; F1: 0.53 vs. 0.56). Including the annotation guidelines in the zero and few-shot prompts obtains worse results (Appendix G, Table 18).

The supervised RoBERTa-based systems trained with other corpora yield mixed results. Circa, SWDA-IA, and Friends-QIA are the ones which outperform the keyword-based baseline (0.53, 0.47 and 0.51 vs. 0.37). We hypothesize that this is due to the fact that BoolQ includes yes-no questions from formal texts (Wikipedia). Unsurprisingly, training with Twitter-YN yields the best results. The question alone is insufficient (F1: 0.29), but combining it with the answer yields better results than the answer alone (0.56 vs. 0.48).

It is observed that Friends-QIA and Circa are the only corpora worth blending with Twitter-YN. Both obtain better results than training with only Twitter-YN (F1: 0.58 in both vs. 0.56). Finally, including in the input the disambiguation texts automatically generated by GPT3 further improves the results (F1: 0.61) blending with Circa. On the other hand, disambiguation texts are detrimental when blending with Friends-QIA. These results were surprising to us as disambiguation texts are often nonsensical. Appendix G (Table 17) provides several examples of disambiguation texts generated by GPT3, including sensical and nonsensical texts.

Regarding statistical significance, it is important to note that blending Circa and using disambiguation texts yields statistically significantly better results than few-shot GPT3. GPT3, however, is by no means useless: it provides a solid few-shot baseline and the disambiguation texts can be used to strengthen the supervised models.

| Error Type | Question | Answer | P,G |
|---|---|---|---|
| *Yes* distractor (4.4%) | Do you own a bandsaw? | Its on my things to acquire list for sure | y,n |
| *No* distractor (28.6%) | Do you like scary movies? | I don't like scary movies I love scary movies | n,y |
| Social media lexicon (20.7%) | Is it foolish to assume that people are always rational? | The keyword is within : #assume 🙌🤔 | u,y |
| Humor (13.2%) | if you haven't fished a fruit fly out of your coffee cup and KEPT DRINKING, do you even like coffee? | But the fruit fly is there to provide the protein. | pn,py |
| Condition or contrast (29.1%) | Do you have a calling in your life? | Well I should hope so, otherwise there'd be birds all over my apartment. | u,py |
| External entities (14.9%) | Is it possible we always have so much road construction here because they built this city on rock and roll? | You must be downtown on the corner of 3rd and Jefferson Starship | u,y |
| Unresponsive (11.4%) | Do you want "old things" to pass away? | You'll always inspire more by your example than your words. | y,u |

Table 6: Most frequent errors made by the best performing system in the *unmatched question and unmatched temporal span* setting (Table 5). Error types may overlap, and P and G stand for *p*rediction and *g*round truth respectively. In addition to *yes* and *no* distractors, social media lexicon (emojis, abbreviation, etc.), humor (including irony and sarcasm), presence of (a) conditions or contrasts or (b) external entities are common errors.

## 6.1 Error Analysis

Table 6 illustrates the most common errors made by our best performing model (RoBERTa blending Twitter-YN (with disambiguation text generated by GPT3) and Circa, Table 5). note that error types may overlap. For example, a wrongly predicted instance may include humor and a *no* distractor).

*Yes* and *no* keywords are not good indicators of answer interpretations (Section 4), and they act as distractors. In the first and second examples, the answers include *sure* and *n't*, but their interpretations are *no* and *yes* respectively. Indeed, we observe them in 33% of errors, especially *no* distractors (28.6%). Many errors (20.7%) contain social media lexicon items such as emojis and hashtags, making their interpretation more difficult. Humor including irony is also a challenge (13.2% of errors). In the example, which also includes a contrast, the answer implies that the author likes coffee as the fruit fly has a positive contribution.

Answers with conditions and contrasts are rarely to be interpreted *unknown*, yet the model often does so. Mentioning external entities also poses a challenge as the model does not have any explicit knowledge about them. Finally, unresponsive answers are always to be interpreted *unknown*, yet the model sometimes (11.4% of errors) fails to properly interpret unresponsive answers and mislabels them as either *yes* or *no*.

## 7 Conclusions

We have presented Twitter-YN, the first corpus of yes-no questions and answers from social media annotated with their interpretations. Importantly, both questions and answer were posted by real users rather than written by crowdworkers on demand. As opposed to traditional question answering, the problem is not to find answers to questions, but rather to *interpret* answers to yes-no questions.

Our analysis shows that answers to yes-no questions vary widely and they rarely include a *yes* or *no* keyword. Even if they do, the underlying interpretation is rarely what one may think the keyword suggests. Answers to yes-no questions usually are long explanations without stating the underlying interpretation (*yes*, *probably yes*, *unknown*, *probably no* or *no*). Experimental results show that the problem is challenging for state-of-the-art models, including large language models such as GPT3.

Our future plans include exploring combination of neural- and knowledge-based approaches. In particular, we believe that commonsense resources such as ConceptNet (Speer et al., 2017), ATOMIC (Sap et al., 2019), and CommonsenseQA (Talmor et al., 2019) may be helpful. The challenge is to differentiate between common or recommended behavior (or commonly held commonsense) and what somebody answered in social media. Going back to the example in Figure 1, oven

manuals and recipes will always instruct people to preheat the oven. The problem is not to obtain a universal ground truth—it does not exist. Instead, the problem is to leverage commonsense and reasoning to reveal nuances about how indirect answers to yes-no questions ought to be interpreted.

## Limitations

Twitter-YN, like any other annotated corpus, is not necessarily representative (and it certainly is not a complete picture) of the problem at hand—in our case, interpreting answers to yes-no questions in user generated content. Twitter-YN may not transfer to other social media platforms. More importantly, our process to select yes-no questions and answers disregards many yes-no questions. In addition to working only with English, we disregard yes-no questions without question marks (e.g., *I wonder what y'all think about preheating the oven*); the questions we work with are limited to those containing four bigrams. We exclude unverified accounts to avoid spam and maximize the chances of finding answers (verified accounts are more popular), but this choice ignores many yes-no questions. Despite the limitations, we believe that the variety of verbs and including two temporal spans alleviates this issue. Regardless of size, no corpus will be complete.

Another potential limitation is the choice of labels. We are inspired by previous work (Section 2 and 3), some of which work with five labels and others with as many as nine. We found no issues using five labels during the annotation effort, but we certainly have no proof that five is the best choice.

Lastly, we report negative results using several prompts and GPT3—although prompting GPT3 is the best approach if zero or few training examples are available. Previous work on prompt engineering and recent advances in large language models makes us believe that it is possible that researchers specializing in prompts may come up with better prompts (and results) than we did. Our effort on prompting and GPT3, however, is an honest and thorough effort to the best of our ability with the objective of empirically finding the optimal prompts. Since these large models learn from random and maliciously designed prompts (Webson and Pavlick, 2022), we acknowledge that other prompts may yield better results.

## Ethics Statement

While tweets are publicly available, users may not be aware that their public tweets can be used for virtually any purpose. Twitter-YN will comply with the Twitter Terms of Use. Twitter-YN will not include any user information, although we acknowledge that it is easily accessible given the tweet id and basic programming skills. The broad temporal spans we work with minimize the opportunities for monitoring users, but it is still possible.

While it is far from our purposes, models to interpret answers to yes-no questions could be used for monitoring individuals and groups of people. Malicious users could post questions about sensitive topics, including politics, health, and deeply held believes. In turn, unsuspecting users may reply with their true opinion, thereby exposing themselves to whatever the malicious users may want to do (e.g., deciding health insurance quotes based on whether somebody enjoys exercising). That said, this hypothetical scenario also opens the door to users to answer whatever they wish, truthful or not—and in the process, mislead the hypothetical monitoring system. The work presented here is about interpreting answers to yes-no questions in Twitter, not figuring the "true" answer. If somebody makes a Twitter persona, we make no attempt to reveal the real person underneath.

## Acknowledgments

This material is based upon work supported by the National Science Foundation under Grant No. 1845757. Any opinions, findings, and conclusions or recommendations expressed in this material are those of the authors and do not necessarily reflect the views of the NSF. We would like to acknowledge the Chameleon platform (Keahey et al., 2020) for providing computational resources.

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

# A  Annotation Guidelines

We give four annotators the following annotation guidelines for each of the five labels. These guidelines are also used in some of the zero-shot and few-shot prompts.

1. *yes* (y)

   The reply is an affirmative reply to the question without reservations.

   Or, the reply states the question in affirmative terms. For example, *Mexican food is delicious.* states *Do you like Mexican food?* in positive terms.

2. *probably yes* (py)

   The reply is an affirmative reply under certain conditions.

   Or, the reply states the question in hopeful terms. For example, *I am keen on eating tacos this weekend* states *Do you like Mexican food?* in hopeful terms.

   Or, the reply is affirmative but refers to a point

of time in the future.

Or, the reply states that an entity (person, group of people, place, organization, etc.) leans towards *yes*. For example, *Q: Do you like Mexican food? A: My whole family loves it.*

3. *no* (n)

The reply is a negative reply to the question without reservations.

Or, the reply states the question in negative terms. For example, *Mexican food is bland* states *Do you like Mexican food?* in negative terms.

4. *probably no* (pn)

The reply is a negative reply under certain conditions.

Or, the reply states that an entity (person, group of people, place, organization, etc.) leans towards no. For example, *Q: Do you like Mexican food? A: My friends avoid Mexican restaurants.*

5. *unknown* (uk)

A reply that is responsive to the question but does not lean towards *yes* or *no*.

Or, a reply that does not address the question at all.

## B   Data Statement

As per the recommendations by Bender and Friedman (2018), we provide here a Data Statement for better understanding of Twitter-YN.

### B.1   Curation Rationale

We develop Twitter-YN to build and evaluate models to interpret answers to yes-no questions in user-generated content. Twitter-YN includes yes-no questions and answers from Twitter along with annotations indicating the interpretation of the answers.

Questions come from main tweets (i.e., any tweet except replies and retweets). Answers are reply tweets to the question tweets (retweets do not count as replies). Questions and answers are selected based on manually defined rules. These rules are developed and refined through an iterative process. We scrape Twitter for question-answer pairs and assign 100 instances to four annotators (each annotator gets the same instances). Based on (a) feedback by each of these four annotators on the questions and answers and (b) agreement scores (linearly weighted Cohen's $\kappa$), the rules are refined.

A new set of 100 random instances is then collected based on the refined rules. This is once again given for annotation to the four annotators. This process continues until annotators find that the questions and answers identified with the rules are actual yes-no questions and answers to the questions.

The final version of the rules used to scrape yes-no questions and answers from Twitter are detailed in Section 3.1. Section 3.2 provides descriptions of the five interpretations annotators choose from (i.e., annotation guidelines), and Table 1 presents examples. All question-answer pairs (4,442) were annotated by two annotators independently. Four annotators participated in the annotations. Inter-annotator agreement (linearly weighted Cohen's $\kappa$) is 0.68, indicating substantial agreement (Artstein and Poesio, 2008); over 0.8 would be nearly perfect. After the independent double annotation process, the adjudicator resolves the disagreements to create the ground truth.

### B.2   Language Variety

The data collection process was carried out from May to July 2022. Question tweets are in English ('en' as per the Twitter API). We also use SpaCy to confirm that the reply is in English. Information on the specific type of English (American, British, Australian etc.) is not available.

### B.3   Speaker Demographic

The questions come from 1,200 unique question tweets posted by verified twitter accounts (as of July 2022). We look specifically at verified users only, since such tweets are more popular and are therefore more likely to have replies. We do not require replies to come from verified accounts. As per Twitter age restrictions, the minimum user age is 13 years for authors of both questions and answers. Speakers are not reachable in our scenario, thus demographic information is limited.

### B.4   Annotator Demographic

Four annotators (including one adjudicator) are part of the annotation process and development of annotation guidelines. All of the individuals are graduate students who are fluent in English. Annotators are three men and one woman, and their ages range from 18 to 40 years old. Ethnic backgrounds are as follows: one Asian and three South Asian. Annotators reported that they are from a middle class economic background.

The final version of the annotation guidelines is developed by considering the annotator feedback during pilot annotations. There is no overlap between the questions and answers used in the pilot annotations and the ones that are in Twitter-YN.

### B.5 Speaker Situation

Text in Twitter-YN is retrieved from Twitter between May and July of 2022. Modality of text is written (typed by speaker). Twitter allows users (speakers) to edit what they tweet. We use the version of the tweet available as of July 2022 regardless of the original posting date. Twitter-YN includes asynchronous interactions since reply tweets are posted only after the tweet with the question is posted. Questions and answers cannot be appear in Twitter simultaneously. The intended audience for the text could be anyone on the Internet.

### B.6 Text Characteristics

Genre of text is in the social media domain. Question-answer instances are generally based on topics such as general advice, day-to-day chores, health, food, finance, music, policies, and politics. The text is informal. Social media cues such as shorter forms of phrases (e.g., lol: laughing out loud), slang, and emojis are common. Most tweets are text only (including Unicode symbols such as emojis), although a few also contain images. Twitter-YN does not include these images.

### B.7 Recording Quality

N/A

### B.8 Other

N/A

### B.9 Provenance Appendix

N/A

## C Results with the other Two Experimental Settings

Tables 7 and 8 present results with the *unmatched question and matched temporal span* and *matched question and matched temporal span* settings. These tables complement the results discussed in Section 5. While valid, these settings are unrealistic in terms of what the model is trained and evaluated with. In the first setting, models are trained with question-answer pairs posted during the same temporal span than those in the test split. In the second setting, models are trained with answers to the same questions seen during training.

Since the test splits are not the same, the results in Tables 7 and 8 are not comparable. Because of the same reason, the results in Table 5 and either Table 7 or 8 cannot be compared either.

## D Hyperparameters

All RoBERTa-based classifiers were tuned using the train and validation splits. Results (Tables 5 and 7–8, 12–14), were obtained with the test split after the tuning process. We experimented with the following ranges for hyperparameters, and chose the optimal values based on the loss calculated with the validation split:

- Learning rate: 1e-5, 2e-5, 3e-5
- Epochs: Up to 200 with early stopping (patience = 3, min_delta = 0.01)
- Batch size: 16, 32
- Blending factor ($\alpha$): 0.2–0.8

Tables 9–11 present the tuned hyperparameters in the three experimental settings. These tables complement the results presented in Tables 5, 7, and 8. Hyperparameters were tuned with the train and validation splits.

## E Results Pretraining with Related Corpora

Tables 12–14 present results pretraining with yes-no questions and then fine-tuning with Twitter-YN rather than blending. These tables complement Tables 5, 7, and 8. Although pretraining sometimes outperforms blending in two settings, blending always outperforms pretraining in the most important setting: *unmatched question and unmatched temporal span*.

## F Detailed Error Analysis

Tables 15 and 16 detail the error types exemplified in Table 6. Specifically, we provide the percentages of errors for each combination of ground truth (gold) and predicted interpretations.

## G Experiments Using Prompts and GPT3

Given the success of of large language models and prompt engineering in many tasks (Brown et al., 2020), we are keen to see whether they are successful at interpreting answers to yes-no questions from Twitter. We experiment with various prompt-based approaches and GPT3. Specifically,

| | All | | | | yes | no | pyes | pno | unk |
|---|---|---|---|---|---|---|---|---|---|
| | Accuracy | P | R | F1 | F1 | F1 | F1 | F1 | F1 |
| Baseline, Majority Class | 0.41 | 0.17 | 0.41 | 0.24 | 0.58 | 0.00 | 0.00 | 0.00 | 0.00 |
| Baseline, Keyword-Based Rules | 0.38 | 0.55 | 0.38 | 0.39 | 0.38 | 0.49 | 0.19 | 0.25 | 0.35 |
|    with negation cue detector | 0.32 | 0.50 | 0.32 | 0.30 | 0.23 | 0.44 | 0.12 | 0.29 | 0.31 |
| RoBERTa trained with yes-no questions corpora | | | | | | | | | |
|   BoolQ | 0.44 | 0.36 | 0.44 | 0.32 | 0.59 | 0.29 | 0.00 | 0.00 | 0.00 |
|   Circa | 0.58 | 0.56 | 0.58 | 0.54 | 0.70 | 0.61 | 0.38 | 0.10 | 0.26 |
|   SwDA-IA | 0.53 | 0.45 | 0.53 | 0.44 | 0.65 | 0.56 | 0.00 | 0.00 | 0.09 |
|   Friends-QIA | 0.56 | 0.53 | 0.56 | 0.53 | 0.71 | 0.58 | 0.30 | 0.00 | 0.28 |
|   Twitter-YN (our corpus) | | | | | | | | | |
|     Question only | 0.41 | 0.27 | 0.41 | 0.30 | 0.57 | 0.24 | 0.00 | 0.00 | 0.00 |
|     Answer only | 0.53 | 0.47 | 0.53 | 0.47 | 0.64 | 0.59 | 0.00 | 0.00 | 0.25 |
|     Question and Answer | 0.58 | 0.54 | 0.58 | 0.56 | 0.70 | 0.60 | 0.23 | 0.00 | 0.45 |
| RoBERTa blending Twitter-YN (question and answer) and other yes-no questions corpora | | | | | | | | | |
|   BoolQ ($\alpha = 0.5$) | 0.60 | 0.61 | 0.60 | **0.60**∗ | 0.71 | 0.62 | 0.37 | 0.25 | 0.52 |
|   Circa ($\alpha = 0.2$) | 0.61 | 0.58 | 0.61 | 0.58 | 0.72 | 0.67 | 0.30 | 0.14 | 0.38 |
|   SWDA-IA ($\alpha = 0.5$) | 0.58 | 0.56 | 0.58 | 0.57 | 0.68 | 0.63 | 0.26 | 0.22 | 0.45 |
|   Friends-QIA ($\alpha = 0.8$) | 0.57 | 0.57 | 0.57 | 0.57 | 0.70 | 0.62 | 0.31 | 0.18 | 0.40 |

Table 7: Results with the test split of Twitter-YN (our corpus) in the *unmatched question and matched temporal span* setting. We present results with (a) baselines, (b) training with yes-no questions corpora (other corpora and the training split of Twitter-YN), and (c) blending the training split of Twitter-YN and other corpora. While direct comparison is unsound because the training and test splits differ, we observe either the same results or just slightly higher than the ones in the *unmatched question and unmatched temporal span* setting (Table 5), especially when blending (last block). In other words, training with questions and answers of the new temporal span does not present an advantage. Statistical significance with respect to training RoBERTa trained with Twitter-YN (Question and Answer, second block) is indicated with and asterisk (McNemar's Test (McNemar, 1947), $p < 0.05$)

| | All | | | | yes | no | pyes | pno | unk |
|---|---|---|---|---|---|---|---|---|---|
| | Accuracy | P | R | F1 | F1 | F1 | F1 | F1 | F1 |
| Baseline, Majority Class | 0.40 | 0.18 | 0.40 | 0.23 | 0.57 | 0.00 | 0.00 | 0.00 | 0.00 |
| Baseline, Keyword-Based Rules | 0.34 | 0.49 | 0.34 | 0.34 | 0.35 | 0.45 | 0.19 | 0.16 | 0.29 |
|    with negation cue detector | 0.29 | 0.52 | 0.29 | 0.29 | 0.23 | 0.45 | 0.15 | 0.17 | 0.27 |
| RoBERTa trained with yes-no questions corpora | | | | | | | | | |
|   BoolQ | 0.50 | 0.36 | 0.50 | 0.40 | 0.63 | 0.49 | 0.00 | 0.00 | 0.00 |
|   Circa | 0.55 | 0.52 | 0.55 | 0.51 | 0.67 | 0.58 | 0.31 | 0.00 | 0.25 |
|   SwDA-IA | 0.52 | 0.60 | 0.52 | 0.52 | 0.68 | 0.59 | 0.20 | 0.13 | 0.26 |
|   Friends-QIA | 0.56 | 0.55 | 0.56 | 0.54 | 0.65 | 0.62 | 0.34 | 0.00 | 0.42 |
|   Twitter-YN (our corpus) | | | | | | | | | |
|     Question only | 0.50 | 0.40 | 0.50 | 0.43 | 0.63 | 0.50 | 0.00 | 0.00 | 0.18 |
|     Answer only | 0.56 | 0.57 | 0.56 | 0.53 | 0.67 | 0.58 | 0.33 | 0.18 | 0.32 |
|     Question and Answer | 0.63 | 0.74 | 0.63 | 0.60 | 0.71 | 0.66 | 0.41 | 0.13 | 0.47 |
| RoBERTa blending Twitter-YN (question and answer) and other yes-no questions corpora | | | | | | | | | |
|   BoolQ ($\alpha = 0.5$) | 0.66 | 0.66 | 0.66 | **0.64** | 0.75 | 0.69 | 0.40 | 0.23 | 0.56 |
|   Circa ($\alpha = 0.5$) | 0.64 | 0.61 | 0.64 | 0.62 | 0.76 | 0.70 | 0.32 | 0.14 | 0.45 |
|   SWDA-IA ($\alpha = 0.5$) | 0.64 | 0.62 | 0.64 | 0.62 | 0.76 | 0.67 | 0.38 | 0.13 | 0.46 |
|   Friends-QIA ($\alpha = 0.8$) | 0.65 | 0.62 | 0.65 | 0.63 | 0.74 | 0.71 | 0.36 | 0.06 | 0.56 |

Table 8: Results with the test split of Twitter-YN (our corpus) in the *matched question and matched temporal span* setting. We present results with (a) baselines, (b) training with yes-no questions corpora (other corpora and the training split of Twitter-YN), and (c) blending the training split of Twitter-YN and other corpora. While direct comparison is unsound because the training and test splits differ, we generally observe higher or the same results than the ones in the *unmatched question and unmatched temporal span* setting (Table 5). Unsurprisingly, using (different) answers to the same questions in training and testing is beneficial. Note, however, that these results are unrealistic.

| Experiment | Learning Rate | Epochs | Batch Size |
|---|---|---|---|
| RoBERTa trained with yes-no questions corpora | | | |
| BoolQ | 3e-5 | 1 | 16 |
| Circa | 3e-5 | 3 | 16 |
| SWDA-IA | 2e-5 | 3 | 16 |
| Friends-QIA | 2e-5 | 2 | 16 |
| Twitter-YN (our corpus) | | | |
|   Question | 1e-5 | 2 | 16 |
|   Answer | 3e-5 | 2 | 16 |
|   Questions and Answer | 2e-5 | 2 | 16 |
| RoBERTa pretraining with yes-no question corpora + finetuning with Twitter-YN | | | |
| BoolQ | 3e-5 | 1+1 | 16 |
| Circa | 3e-5 | 3+1 | 16 |
| SWDA-IA | 2e-5 | 3+1 | 16 |
| Friends-QIA | 2e-5 | 2+1 | 16 |
| RoBERTa blending Twitter-YN (question and answer) and other yes-no questions corpora | | | |
| BoolQ | 3e-5 | 1 | 16 |
| Circa | 2e-5 | 1 | 16 |
|   +disambiguation text (GPT3) | 2e-5 | 2 | 16 |
| SWDA-IA | 1e-5 | 2 | 16 |
| Friends-QIA | 2e-5 | 1 | 16 |
|   +disambiguation text (GPT3) | 2e-5 | 2 | 16 |

Table 9: Hyperparameters with the *unmatched question and unmatched temporal span* setting found after tuning with the development set. This table complements Table 5.

| Experiment | Learning Rate | Epochs | Batch Size |
|---|---|---|---|
| RoBERTa trained with yes-no questions corpora | | | |
| BoolQ | 1e-5 | 1 | 16 |
| Circa | 3e-5 | 1 | 16 |
| SWDA-IA | 3e-5 | 2 | 16 |
| Friends-QIA | 3e-5 | 3 | 16 |
| Twitter-YN (our corpus) | | | |
|   Question | 1e-5 | 1 | 16 |
|   Answer | 3e-5 | 2 | 16 |
|   Question and Answer | 2e-5 | 2 | 16 |
| RoBERTa pretraining with yes-no question corpora + finetuning with Twitter-YN | | | |
| BoolQ | 1e-5 | 1+2 | 16 |
| Circa | 3e-5 | 1+1 | 16 |
| SWDA-IA | 3e-5 | 2+2 | 16 |
| Friends-QIA | 3e-5 | 3+1 | 16 |
| RoBERTa blending Twitter-YN (question and answer) and other yes-no questions corpora | | | |
| BoolQ | 2e-5 | 3 | 16 |
| Circa | 3e-5 | 3 | 16 |
| SWDA-IA | 1e-5 | 3 | 16 |
| Friends-QIA | 2e-5 | 2 | 16 |

Table 10: Hyperparameters with the *unmatched question and matched temporal span* setting found after tuning with the development set. This table complements Table 7.

| Experiment | Learning Rate | Epochs | Batch Size |
|---|---|---|---|
| RoBERTa trained with yes-no questions corpora | | | |
| BoolQ | 1e-5 | 3 | 16 |
| Circa | 3e-5 | 1 | 16 |
| SWDA-IA | 2e-5 | 3 | 16 |
| Friends-QIA | 3e-5 | 3 | 16 |
| Twitter-YN (our corpus) | | | |
|   Question | 2e-5 | 3 | 16 |
|   Answer | 3e-5 | 2 | 16 |
|   Question and Answer | 2e-5 | 2 | 16 |
| RoBERTa pretraining with yes-no question corpora + finetuning with Twitter-YN | | | |
| BoolQ | 1e-5 | 3+3 | 16 |
| Circa | 3e-5 | 1+2 | 16 |
| SWDA-IA | 2e-5 | 3+2 | 16 |
| Friends-QIA | 3e-5 | 3+1 | 16 |
| RoBERTa blending Twitter-YN (question and answer) and other yes-no questions corpora | | | |
| BoolQ | 1e-5 | 3 | 16 |
| Circa | 2e-5 | 2 | 16 |
| SWDA-IA | 1e-5 | 2 | 16 |
| Friends-QIA | 2e-5 | 2 | 16 |

Table 11: Hyperparameters with the *matched question and matched temporal span* setting found after tuning with the development set. This table complements Table 8.

| | Overall | | | | yes | no | pyes | pno | unk |
|---|---|---|---|---|---|---|---|---|---|
| | Accuracy | P | R | F1 | F1 | F1 | F1 | F1 | F1 |
| BoolQ | 0.51 | 0.54 | 0.51 | 0.41 | 0.64 | 0.57 | 0.03 | 0.00 | 0.02 |
| Circa | 0.60 | 0.60 | 0.60 | 0.58 | 0.71 | 0.62 | 0.38 | 0.22 | 0.45 |
| SwDA-IA | 0.55 | 0.55 | 0.55 | 0.51 | 0.69 | 0.55 | 0.36 | 0.00 | 0.33 |
| Friends-QIA | 0.58 | 0.55 | 0.58 | 0.54 | 0.70 | 0.60 | 0.26 | 0.00 | 0.42 |

Table 12: Results with RoBERTa pretrained on relevant yes-no questions corpora and fine-tuned on Twitter-YN in the *unmatched question and unmatched temporal span* setting. Pretraining obtains worse results than blending (Table 5).

| | Overall | | | | yes | no | pyes | pno | unk |
|---|---|---|---|---|---|---|---|---|---|
| | Accuracy | P | R | F1 | F1 | F1 | F1 | F1 | F1 |
| BoolQ | 0.59 | 0.51 | 0.59 | 0.54 | 0.72 | 0.62 | 0.00 | 0.00 | 0.42 |
| Circa | 0.61 | 0.63 | 0.61 | 0.59 | 0.74 | 0.63 | 0.25 | 0.06 | 0.47 |
| SwDA-IA | 0.57 | 0.49 | 0.57 | 0.51 | 0.71 | 0.59 | 0.00 | 0.00 | 0.33 |
| Friends-QIA | 0.57 | 0.55 | 0.57 | 0.51 | 0.69 | 0.59 | 0.23 | 0.00 | 0.25 |

Table 13: Results with RoBERTa pretrained on relevant yes-no questions corpora and fine-tuned on Twitter-YN in the *unmatched question and matched temporal span* setting.

| | Overall | | | | yes | no | pyes | pno | unk |
|---|---|---|---|---|---|---|---|---|---|
| | Accuracy | P | R | F1 | F1 | F1 | F1 | F1 | F1 |
| BoolQ | 0.59 | 0.56 | 0.59 | 0.56 | 0.70 | 0.62 | 0.19 | 0.00 | 0.48 |
| Circa | 0.64 | 0.62 | 0.64 | 0.62 | 0.72 | 0.71 | 0.36 | 0.18 | 0.47 |
| SwDA-IA | 0.62 | 0.64 | 0.62 | 0.61 | 0.73 | 0.69 | 0.39 | 0.00 | 0.50 |
| Friends-QIA | 0.61 | 0.64 | 0.61 | 0.60 | 0.71 | 0.67 | 0.40 | 0.06 | 0.50 |

Table 14: Results with RoBERTa pretrained on relevant yes-no questions corpora and fine-tuned on Twitter-YN in the *matched question and matched temporal span* setting.

| Error Type | Ground Truth | Predicted | Percentage |
|---|---|---|---|
| *Yes* distractor | no | yes | 0.88 |
| | yes | no | 0.88 |
| | probably yes | yes | 0.44 |
| | yes | probably yes | 0.44 |
| | unknown | no | 0.44 |
| | no | unknown | 0.44 |
| | no | probably yes | 0.44 |
| | yes | unknown | 0.44 |
| *No* distractor | unknown | no | 5.29 |
| | yes | unknown | 5.29 |
| | yes | no | 3.52 |
| | probably yes | no | 3.08 |
| | probably no | no | 2.20 |
| | probably yes | yes | 1.76 |
| | no | unknown | 0.88 |
| | yes | probably yes | 0.88 |
| | yes | probably no | 0.88 |
| | no | yes | 0.88 |
| | unknown | yes | 0.88 |
| | probably yes | probably no | 0.88 |
| | probably no | unknown | 0.44 |
| | probably no | probably yes | 0.44 |
| | unknown | probably no | 0.44 |
| | probably yes | unknown | 0.44 |
| | unknown | probably yes | 0.44 |
| External Entities | yes | unknown | 3.52 |
| | probably yes | unknown | 2.64 |
| | probably yes | yes | 2.20 |
| | probably yes | no | 1.32 |
| | no | unknown | 0.88 |
| | unknown | yes | 0.88 |
| | unknown | no | 0.88 |
| | yes | no | 0.88 |
| | no | yes | 0.44 |
| | probably no | no | 0.44 |
| | probably no | yes | 0.44 |
| | yes | probably yes | 0.44 |

Table 15: Detailed error analysis showing the ground truth and predicted labels of the best model (Part 1). This table complements Table 6.

| Error Type | Ground Truth | Predicted | Percentage |
|---|---|---|---|
| Social media lexicon | yes | unknown | 5.29 |
| | probably yes | unknown | 3.08 |
| | no | yes | 2.20 |
| | unknown | no | 1.32 |
| | unknown | probably yes | 1.32 |
| | probably no | unknown | 1.32 |
| | probably yes | yes | 1.32 |
| | probably no | no | 0.88 |
| | no | unknown | 0.88 |
| | probably yes | no | 0.88 |
| | probably no | probably yes | 0.44 |
| | yes | probably no | 0.44 |
| | unknown | yes | 0.44 |
| | no | probably no | 0.44 |
| | yes | probably yes | 0.44 |
| Humor | no | unknown | 2.64 |
| | yes | no | 2.20 |
| | yes | unknown | 1.76 |
| | no | yes | 1.32 |
| | probably yes | yes | 0.88 |
| | probably no | no | 0.88 |
| | probably no | unknown | 0.88 |
| | unknown | probably yes | 0.44 |
| | no | probably yes | 0.44 |
| | yes | probably yes | 0.44 |
| | probably no | yes | 0.44 |
| | probably yes | unknown | 0.44 |
| | probably no | probably yes | 0.44 |
| Condition or contrast | no | unknown | 3.52 |
| | probably yes | yes | 3.52 |
| | no | yes | 3.08 |
| | probably no | no | 2.64 |
| | yes | probably yes | 2.64 |
| | yes | unknown | 2.64 |
| | no | probably no | 1.76 |
| | unknown | no | 1.76 |
| | probably yes | unknown | 1.32 |
| | yes | no | 1.32 |
| | unknown | probably yes | 0.88 |
| | probably no | yes | 0.88 |
| | no | probably yes | 0.88 |
| | probably no | probably yes | 0.44 |
| | probably yes | no | 0.44 |
| | unknown | yes | 0.44 |
| | probably no | unknown | 0.44 |
| | probably yes | probably no | 0.44 |
| Unresponsive | unknown | no | 6.16 |
| | unknown | yes | 3.52 |
| | unknown | probably yes | 1.76 |

Table 16: Detailed error analysis showing the ground truth and predicted labels of the best model (Part 2). This table complements Table 6.

we experiment with GPT3's *text-davinci-002* engine, *temperature=0.7*, *frequency_penalty=0*, *presence_penalty=0* and *top_p=1*. Our prompts do not exactly follow previous work, but they are inspired by previous work (Liu et al., 2021):

**Zero-Shot.** We use prefix prompts (Li and Liang, 2021; Lester et al., 2021) since such prompts work well with generative language models such as GPT3 (Liu et al., 2021). As shown by Efrat and Levy, 2020, we also experiment with the same zero-shot prompt including the annotation guidelines. Specifically, we experiment with two zero-shot prompts:

- The first zero-shot prompt includes (a) the yes-no question and answer, and (b) a plain English question asking GPT3 to interpret the answer: *What does the given answer mean? Choose only from the following options: yes, probably yes, no, probably no, unknown.* Figure 4 illustrates this prompt.
- The second zero-shot prompt includes the annotation guidelines (Appendix A) for each label, and then the same content as the first prompt (i.e., yes-no question and answer followed by the plain English question asking GPT3 to interpret the answer). Figure 5 illustrates this prompt.

**Few-Shot.** We experiment with longer versions of the zero-shot prompts that include two examples from the training split per label. Specifically, we add two examples per label (total: 10 examples) to create few-shot versions of the zero-shot prompts detailed above. Figures 6 and 7 illustrate the few-shot prompts (with and without guidelines respectively).

**Prompt-Derived Knowledge.** Liu et al. (2022) have shown that integrating knowledge derived from GPT3 in question-answering models is beneficial. We follow a similar approach by prompting GPT3 to generate a *disambiguation text* given a question-answer pair from Twitter-YN. Then we complement the input to the RoBERTa-based classifier with the disambiguation text. Figure 8 illustrated this prompt, and Table 17 provides examples of disambiguation texts. As the table shows, certain disambiguation texts are invalid.

### G.1 Results

Table 18 presents the results using GPT3. The best RoBERTa-based system obtains an overall F1 of 0.58 (Table 5). All the systems using GPT3 obtain F1s under 0.58. Including the disambiguation text to RoBERTa is detrimental in case of Friends-QIA (F1: 0.56) but beneficial in case of Circa (F1: 0.61). Unsurprisingly, few-shot approaches obtain better results than zero-shot prompts (0.53, 0.50 vs. 0.46, 0.41). Perhaps surprisingly, including the annotation guidelines is detrimental (zero-shot: 0.46 vs. 0.41, few-shot: 0.53 vs. 0.50).

We acknowledge that prompt engineering may lead to better results with GPT3. We refer the reader to the Limitations section for a discussion.

```
Question: <yes-no question tweet>
Answer: <reply tweet>
What does the given answer mean? Choose only from the following options:
yes, probably yes, no, probably no, unknown.
<answer space>
```

Figure 4: Template to generate zero-shot prompts for GPT3

```
yes:
[guidelines for yes]

probably yes:
[guidelines for probably yes]

no:
[guidelines for no]

probably no:
[guidelines for probably no]

unknown:
[guidelines for unknown]

Question: <yes-no question tweet>
Answer: <reply tweet>
What does the given answer mean? Choose only from the following
options: yes, probably yes, no, probably no, unknown.
<answer space>
```

Figure 5: Template to generate zero-shot prompts with annotation guidelines for GPT3. We refer the reader to Appendix A for the annotation guidelines for each class.

```
Question: Do you have any weird things that you're really really
good at?
Answer: i can balance things, like tennis rackets or golf clubs,
on my nose.
The given answer can be interpreted as yes.

Question: Do you guys believe in ghosts?
Answer: i do believe in ghost and i'm not afraid of ghost.
The given answer can be interpreted as yes.

Question: Do you think that inmates should get access to free
university education while in prison?
Answer: if our intent is to rehabilitate, we should absolutely
offer higher education to inmates.
The given answer can be interpreted as probably yes.

Question: Do you like cereal?
Answer: sometimes, i have to be in the mood for it.
The given answer can be interpreted as probably yes.

Question: Is it theft to take the pen from your hotel room?
Answer: It's in the marketing budget. They want you to take the
swag.
The given answer can be interpreted as no.

Question: Is it too early for pizza?
Answer: every time of day can be pizza time.
The given answer can be interpreted as no.

Question: On a work zoom, is it unprofessional to obviously be in
your bed?
Answer: anyone who would be working with you needs to know you're
a bed boy.
The given answer can be interpreted as probably no.

Question: Is it a crime to get extra vaccination shots?
Answer: we'll all be doing it for the rest of our lives so i
hope not.
The given answer can be interpreted as probably no.

Question: Do you pee in the ocean?
Answer: tell gio to cut his hair, he looks like an idiot.
The given answer can be interpreted as unknown.

Question: Do you believe your phone is listening to you?
Answer: even if it does, i don't mind. i've got nothing to hide.
The given answer can be interpreted as unknown.

Question: <yes-no question tweet>
Answer: <reply-tweet>
Choose the correct interpretation: yes, no, probably yes,
probably no, unknown.
<answer space>
```

Figure 6: Template to generate few-shot prompts for GPT3

```
yes:
[guidelines for yes]

probably yes:
[guidelines for probably yes]

no:
[guidelines for no]

probably no:
[guidelines for probably no]

unknown:
[guidelines for unknown]

Examples:

[example1 for yes]
[example2 for yes]

[example1 for probably yes]
[example2 for probably yes]

[example1 for no]
[example2 for no]

[example1 for probably no]
[example2 for probably no]

[example1 for unknown]
[example2 for unknown]

Question: <yes-no question tweet>
Answer: <reply tweet>
What does the given answer mean? Choose only from the following
options: yes, probably yes, no, probably no, unknown.
<answer space>
```

Figure 7: Template to generate few-shot prompts with annotation guidelines for GPT3. We refer the reader to Appendix A for the annotation guidelines for each class, and Figure 6 for examples.

```
Question: Do you have any weird things that you're
really really good at?
Answer: i can balance things, like tennis rackets or
golf clubs, on my nose.
The given answer can be interpreted as "yes", because the
answerer mentions a list of weird things he or she is
really good at.

Question: <yes-no question tweet>
Answer: <reply tweet>
<answer space>
```

Figure 8: Template to generate the prompt to elicit knowledge (or disambiguation text) regarding the interpretation of an answer with GPT3 (i.e., a justification for the interpretation). The prompt includes an example of disambiguation text (e.g., *the answerer mentions a list of weird things he or she is really good at*). GPT3 always provides answers following the template *The given answer can be interpret as [*label*], because the answerer [*disambiguation text*]*. We use the disambiguation text as an additional input for the RoBERTa-based classifier.

| Question | Answer | G. | Disambiguation (from GPT3) | Valid? |
|---|---|---|---|---|
| My name regularly gets auto-corrected to "Satanism". Get a surprising number of messages addressed to it. Do you have an autocorrect hell? | An old phone of mine used to change yorkshire to workshy. outrageous | y | Does not mention having an autocorrect hell | False |
| Do you still trust anything the media says? | Modern media can make a collective lie to become the truth | n | Mentions how media can manipulate the truth, they do not trust anything the media says | True |
| Do you still super follow me? | I had to chose between spotify and superfollowing | uk | Chose spotify over superfollowing the asker | False |
| Do you have the courage to be disliked? | I am not able to answer this question. | pn | Admits they cannot answer the question, indicating they do not have the courage to be disliked | True |
| Do you bring your phone while using the bathroom? | Depends how long I think I will be there! | py | Implies that they do not bring their phone while using the bathroom | False |

Table 17: Disambiguation texts produced by GPT3. 'G' indicates the gold label. 'Valid?' indicates whether the disambiguation is valid based on the gold label. In the first example, the answer mentions an "auto correct hell," however, the disambiguation disregards this critical term. In the second and fourth examples, the answer is correctly interpreted by GPT3 by specifying media manipulation and capturing the fact that not answering the question shows lack of courage respectively. In the third and fifth examples, GPT3 incorrectly interprets that the author of the answer prefers Spotify and does not bring a phone in the bathroom respectively. These examples show that GPT3 sometimes produces invalid disambiguation texts. As Table 5 shows, the disambiguation texts can be both beneficial and detrimental.

| | All | | | | yes | no | pyes | pno | unk |
|---|---|---|---|---|---|---|---|---|---|
| | Accuracy | P | R | F1 | F1 | F1 | F1 | F1 | F1 |
| GPT3, Zero-shot | 0.44 | 0.55 | 0.44 | 0.46 | 0.56 | 0.51 | 0.16 | 0.13 | 0.43 |
|   + annotation guidelines | 0.42 | 0.55 | 0.42 | 0.41 | 0.51 | 0.48 | 0.12 | 0.00 | 0.40 |
| GPT3, Few-shot | 0.54 | 0.55 | 0.54 | 0.53 | 0.67 | 0.55 | 0.14 | 0.30 | 0.49 |
|   + annotation guidelines | 0.52 | 0.57 | 0.52 | 0.50 | 0.66 | 0.50 | 0.15 | 0.16 | 0.48 |
| RoBERTa blending Twitter-YN (question and answer) and other yes-no questions corpora | | | | | | | | | |
|   Circa (Table 5) | 0.60 | 0.59 | 0.60 | **0.58**† | 0.72 | 0.65 | 0.37 | 0.25 | 0.44 |
|     + disambiguation text (GPT3) | 0.62 | 0.62 | 0.62 | **0.61**† | 0.75 | 0.67 | 0.35 | 0.22 | 0.57 |
|   Friends-QIA (Table 5) | 0.60 | 0.59 | 0.60 | **0.58**† | 0.71 | 0.63 | 0.38 | 0.20 | 0.45 |
|     + disambiguation text (GPT3) | 0.56 | 0.56 | 0.56 | 0.56 | 0.70 | 0.63 | 0.36 | 0.24 | 0.37 |

Table 18: Results obtained with GPT3 in the *unmatched question and unmatched temporal span* scenario. These results are moderately worse than the ones obtained with RoBERTa-based classifiers (Table 5), although the zero-shot prompts obtain results better than the baselines without requiring any annotations. Statistical significance with respect to few-shot GPT3 (first block) is indicated with an dagger (†) (McNemar's Test, $p < 0.05$).