# OpenReview forum: "Interpreting Answers to Yes-No Questions in User-Generated Content"
_EMNLP/2023/Conference — EMNLP 2023 Findings_

### Official Review · Reviewer_v4HH · 2023-08-06

**Soundness:** 4

**Excitement:**

4: Strong: This paper deepens the understanding of some phenomenon or lowers the barriers to an existing research direction.

**Paper Topic And Main Contributions:**

The authors propose a new yes/no QA corpus for exploring the type of answer (yes, no, probably yes, probably no and unknown). This corpus comprises 4,442 pairs of QA from Twitter (1.200 questions; about 3.7 answers per question). The authors provide a linguistic profile of the corpus. Then, they compare different methods: baselines (majority and rule-based), RoBERTa and GPT3. In addition, they tried fine-tuning with another corpus and then the proposed one, as well as blending the corpora.

**Questions For The Authors:**

Will the corpus be made available? What will be the license of use?

How was the statistical confidence calculated?

**Reasons To Accept:**

The corpus. It follows the standard approaches for data gathering and annotation.

Linguistic profile of yes/no question answering.

The explored models are reasonable and well-described, and the error analysis may help improve the model.

**Reasons To Reject:**

None

**Reproducibility:**

5: Could easily reproduce the results.

**Reviewer Confidence:**

4: Quite sure. I tried to check the important points carefully. It's unlikely, though conceivable, that I missed something that should affect my ratings.

---

> ### Author Rebuttal · Authors · 2023-08-26
>
> Your recommendation of our work to EMNLP 2023 is highly appreciated. We thank you for your positive reviews.
>
> **Questions for the authors**
>
> RE: Will the corpus be made available? What will be the license of use?
> - Yes. Twitter-YN will be made available (as stated in the footnote of page 1). We will follow Twitter Terms of Service and release our annotations using “CC BY.”
>
> RE: How was the statistical confidence calculated?
> - We refer the reviewer to the McNemar’s Test (McNemar, 1947), as cited in the caption for Table 5.

---

### Official Review · Reviewer_heYF · 2023-08-11

**Soundness:** 4

**Excitement:**

4: Strong: This paper deepens the understanding of some phenomenon or lowers the barriers to an existing research direction.

**Paper Topic And Main Contributions:**

This paper presents a dataset that focuses on interpreting answers to yes/no questions in social media. This is challenging because the goal isn’t to search for a keyword within the response, but to interpret it. The paper presents this hand-annotated dataset and experiments with several baselines to showcase that this problem warrants further exploration.


**Reasons To Accept:**

The paper focuses on an interesting problem that even SOTA models don’t seem to be able to do well on. It is an interesting avenue of research and the dataset seems valuable. The methodology of the work is sound, and the authors have clearly made an effort to outline their experimental and dataset-related details in the Appendix.


**Reasons To Reject:**

N/A. I think it is a great paper and it should be presented at EMNLP.

**Reproducibility:**

5: Could easily reproduce the results.

**Reviewer Confidence:**

4: Quite sure. I tried to check the important points carefully. It's unlikely, though conceivable, that I missed something that should affect my ratings.

---

> ### Author Rebuttal · Authors · 2023-08-26
>
> Your recommendation of our work to EMNLP 2023 is highly appreciated. We thank you for your positive reviews.

---

### Official Review · Reviewer_8ozY · 2023-08-12

**Typos Grammar Style And Presentation Improvements:** See "Questions For The Authors"
**Soundness:** 2

**Excitement:**

2: Mediocre: This paper makes marginal contributions (vs non-contemporaneous work), so I would rather not see it in the conference.

**Paper Topic And Main Contributions:**

The paper presents research on understanding yes-no questions that are written freely in the social media domain. The authors introduce a new dataset with over 4,000 question-answer pairs and human-annotated yes-no labels. They also test some basic models to show the task's challenges.



**Questions For The Authors:**

Questions For The Authors

A. Introduction & Motivation: The introduction does not adequately motivate the task. Could you clarify the research gap and whether other works cover it?

B. Line 064: Regarding the qualitative analysis of errors made by your model, what significant conclusions were derived?

C. Line 069: The statement about answers being available is a bit unclear. How did you ascertain that the answers are present?

D. Line 137: What do you mean by "interpreting answers" in this context? Are you classifying answers into yes/no/unknown categories?

E. Section 3: Why didn't you use a classifier trained on BOOLQ yes-no (which identifies questions with high precision) as opposed to a heuristic approach?

F. Line 172 & 182: You mention that avoiding other questions with 'wh' words is crucial for precision. Could you quantify this improvement?

G. Line 174: The repetition of the word "avoid" makes the sentence cumbersome. Consider rephrasing.

H. Line 178: What proportion of questions with numbers pertains to cryptocurrencies or stocks? Have any clustering, topic modelling, or classification methods been applied?

I. Line 201: While the approach you've mentioned is commendable, how did you ensure the prevention of data leakage, especially considering duplicates or paraphrased questions from different years?

J. Line 243: Was there a research ethics agreement in place for recruiting graduate students? What were their demographics, and were they native speakers?

K. Line 270: The title "Are Keywords Enough to Interpret Answers?" seems overstated, given that the keyword list omits obvious examples like "yep" for yes.

L. Line 304: Your analysis indicates that the "me" pronoun is more common with "Yes". Why is this significant or interesting? Could you provide a discussion on this?

M. Line 354: You mention that the results are reported using the test split. Could you specify where these results are reported or provide a direct reference/link within the paper?

N. Line 371: The statement about blending is not clear. Could you clarify what you mean by "many yes-no questions" and where they originate from?

O. GPT Experiments: Can you provide more detailed descriptions of your experiments involving the GPT model?

**Reasons To Accept:**

* The task addressed in the paper is genuine, and the curated dataset can potentially be valuable for both the academic community and industry.
* By evaluating several baseline models, the authors provide valuable insights into the challenges of the task.

**Reasons To Reject:**

* The paper currently reads more like a draft than a polished conference submission. It requires thorough restructuring and proofreading due to unclear sentences, repetitions, and confusing sections.
* The paper fails to convey the significance of the task effectively. There's a lack of context and clear delineation of the research gaps in the introduction. A more comprehensive introduction that outlines the context, importance, and absence of similar resources would benefit a broader audience.
* Several claims in the paper lack supporting evidence. Examples include ambiguous statements about improvements in precision without concrete numbers and criticisms of keyword-based methods based on a limited set of keywords.
* Shallow analysis: The paper provides a shallow linguistic dataset analysis without clear takeaways or conclusions. If the analysis isn't enhancing the paper's value or clarity, it might be better suited for the appendix. Conversely, essential details in the appendix, such as dataset splitting justifications, should be integrated into the main text.
* The methods proposed may need reconsideration, especially given potential issues with dataset splitting.
* The proposed dataset is quite small.

**Reproducibility:**

2: Would be hard pressed to reproduce the results. The contribution depends on data that are simply not available outside the author's institution or consortium; not enough details are provided.

**Reviewer Confidence:**

4: Quite sure. I tried to check the important points carefully. It's unlikely, though conceivable, that I missed something that should affect my ratings.

---

> ### Author Rebuttal · Authors · 2023-08-26
>
> Thank you for the detailed review and questions.
>
> **Reasons to reject**
>
> RE: The paper currently reads more like a draft than a polished conference submission. It requires thorough restructuring and proofreading due to unclear sentences, repetitions, and confusing sections.
> - We would need specifics to address this weakness. The other three reviewers appear to disagree.
>
> RE: The paper fails to convey the significance of the task effectively. There's a lack of context and clear delineation of the research gaps in the introduction. A more comprehensive introduction that outlines the context, importance, and absence of similar resources would benefit a broader audience.
> - Please see Section 1, line 066 onward (Motivation). Additionally, recent previous work (Section 2) supports that the problem is worthwhile. Finally, the results are relatively low even with large language models, demonstrating that the problem is challenging (Sections 6 and 7). Section 2 distinguishes between traditional question answering (including social media, line 127-149), previous work on yes-no questions, and our work.
>
> RE: Several claims in the paper lack supporting evidence. Examples include ambiguous statements about improvements in precision without concrete numbers and criticisms of keyword-based methods based on a limited set of keywords.
> - Assuming “improvements in precision without concrete numbers” refers to line 172 (avoiding questions with wh-words to identify yes-no questions), the precision is 100% in our sample. We did not observe any yes-no questions with a wh-word. Just to clarify, wh-words are who, where, etc. (line 162-163) (and they expect a person, location, etc. for an answer; not a yes or no). Thank you for pointing this out. We will include the number.
>
> - Our goal is to present an unsupervised keyword-based baseline. Any keyword-based method will be based on a limited set of keywords by definition. Additionally, note that many answers that contain yes keywords must be interpreted No and vice versa (Figure 3). Adding more keywords would not bypass this issue. We believe that explaining this point (line 270-285) is an important analysis which helps understand the dataset and difficulty of the problem.
>
> RE: Shallow analysis: The paper provides a shallow linguistic dataset analysis without clear takeaways or conclusions. If the analysis isn't enhancing the paper's value or clarity, it might be better suited for the appendix. Conversely, essential details in the appendix, such as dataset splitting justifications, should be integrated into the main text.
> - We believe it is important to justify the diversity (bigrams, many verbs, etc.) of the new dataset and provide a linguistic analysis (Section 4.1). We acknowledge that the importance of analyses is subjective, but we do argue that the analyses belong to the main paper.
>
> RE: Conversely, essential details in the appendix, such as dataset splitting justifications, should be integrated into the main text.
> - The dataset splits (train / validation / test) using the three settings (matched/unmatched; question/temporal) (and the justification) are described in the main paper (line 328-342). The main paper includes results with the most realistic setting, and Appendix C includes results with the two least realistic settings.
> We are confident this is the right content for the main paper and appendices.
>
> RE: The methods proposed may need reconsideration, especially given potential issues with dataset splitting.
> - The results presented in the paper are for the “unmatched question unmatched temporal span” setting (as stated in line 330-339). As explained there, this means that (by definition) the question-answer pairs in the training and test split (a) do not have any overlapping question and (b) were published in non-overlapping temporal spans.
> Therefore, there are no issues with dataset splitting.
>
> RE: The proposed dataset is quite small.
> - The dataset has 4,442 question-answers pairs (line 006). This is subjective, but we acknowledge the corpus size in the Limitations. No corpus accounts for everything. Further, like everybody else, we are bound by a limited annotation budget. That said, we want to point out that the experimental results show that our models learn from the dataset (Table 5).
>
>
> **Questions for the authors**
>
> A.
> RE: Introduction & Motivation: The introduction does not adequately motivate the task. Could you clarify the research gap and whether other works cover it?
> - Please see Section 1, line 066 onward (Motivation). Additionally, recent previous work (Section 2) supports that the problem is worthwhile. Finally, the results are relatively low even with large language models, demonstrating that the problem is challenging (Sections 6 and 7).
> Section 2 distinguishes between traditional question answering (including social media, line 127-149), previous work on yes-no questions, and our work.
>
> B.
> RE: Line 064: Regarding the qualitative analysis of errors made by your model, what significant conclusions were derived?
> - We describe and exemplify the most frequent error types in Section 6, we refer the reviewer to that section for an answer.
>
> C.
> RE: Line 069: The statement about answers being available is a bit unclear. How did you ascertain that the answers are present?
> - Let us make it more clear: We collect yes-no questions from Twitter. The reply tweets to the tweets containing the yes-no questions are the answers to those questions. So, the answers are available (i.e., they do not need to be retrieved from a document collection). The problem is to figure out the correct interpretation of the answers. Note that generally question answering is about finding the answer to a question (line 066-071).
>
>     We will clarify this; thank you for pointing out your confusion.
>
> D.
> RE: Line 137: What do you mean by "interpreting answers" in this context? Are you classifying answers into yes/no/unknown categories?
> - Yes, as explained in line 027-034 and Figure 1.
>
> E.
> RE: Section 3: Why didn't you use a classifier trained on BOOLQ yes-no (which identifies questions with high precision) as opposed to a heuristic approach?
> - BoolQ contains only yes-no questions, so we would still need negative examples. While your suggestion is valid (and training such a classifier would not be difficult), our heuristic approach requires less effort and is very effective: the annotation process did not identify issues. We also would like to point out that identifying the yes-no questions is not a major contribution; it is only a step towards the creation of our dataset.
>
> F.
> RE: Line 172 & 182: You mention that avoiding other questions with 'wh' words is crucial for precision. Could you quantify this improvement?
> - The precision is 100% in our sample. We did not observe any yes-no questions with a wh-word. Just to clarify, wh-words are who, where, why, etc. (line 162-163) (and they expect a person, location, reason, etc. for an answer; not a yes or no).
>
>     Thank you for the question, we will include the number and reword.
>
> G.
> RE: Line 174: The repetition of the word "avoid" makes the sentence cumbersome. Consider rephrasing.
> - Thank you for helping us improve readability. We will use “discarding tweets with X” and other similar paraphrases to minimize potential confusions.
>
> H.
> RE: Line 178: What proportion of questions with numbers pertains to cryptocurrencies or stocks? Have any clustering, topic modeling, or classification methods been applied?
> - Out of a random sample of 100 instances, 68% of questions with numbers pertained to cryptocurrencies and stocks. The remaining ones covered politics (e.g. number of votes, years) and sports (e.g., scores). We will include these details in the paper. This is a manual analysis. The methods suggested by the reviewer scale but would introduce errors.
>
> I.
> RE: Line 201: While the approach you've mentioned is commendable, how did you ensure the prevention of data leakage, especially considering duplicates or paraphrased questions from different years?
> - By definition (line 330-339), the “unmatched question unmatched temporal span” setting does not have in the training and test splits (a) questions published in overlapping temporal spans or (b) the same questions. Therefore, there is no data leakage.
> While this is not a proof, the results from training with Twitter-YN (question+answer, F1: 0.56) indicate that there is no data leakage, otherwise the F1 would be much higher.
>
> J.
> RE: Line 243: Was there a research ethics agreement in place for recruiting graduate students? What were their demographics, and were they native speakers?
> - We are employed by large universities. The research presented in the paper was done in accordance to our research guidelines and regulations, including ethics. We have mandated training and are highly regulated.
> Please find details about the annotators in the Data Statement (Appendix B). They are not native speakers but have lived in an English-speaking country for years. Let us also point out that not everyone in social media using English is a native speaker, so the benefit of having native speakers as annotators is not obvious to us. Annotators are co-authors as they are a key contributor to the paper.
>
> K.
> RE: Line 270: The title "Are Keywords Enough to Interpret Answers?" seems overstated, given that the keyword list omits obvious examples like "yep" for yes.
> - As stated earlier, no list of keywords is complete. Additionally, note that many answers that contain _yes_ keywords must be interpreted No and vice versa (Figure 3). So more than keyword matching is needed: no matter how long the list of keywords, as stated in the paper ( line 263-269), keyword matching is not enough because keywords are misleading.
>
> L.
> RE: Line 304: Your analysis indicates that the "me" pronoun is more common with "Yes". Why is this significant or interesting? Could you provide a discussion on this?
> - This is an empirical finding. We did not anticipate this finding. We hesitate to make strong claims, but the field of Psychology has made very interesting insights about pronoun usage, in particular first person pronouns (see, for example, the work by James Pennenbaker).
>
> M.
> RE: Line 354: You mention that the results are reported using the test split. Could you specify where these results are reported or provide a direct reference/link within the paper?
> - “[...] we report results with the test split” means that all the results in the paper (Table 5 and appendices) are the results obtained with the test split (after training and validating with the training and validation splits).
>
> N. Line 371:
> RE: The statement about blending is not clear. Could you clarify what you mean by "many yes-no questions" and where they originate from?
> - Thank you, we will clarify this. “Many” here refers to the data coming from related corpora and Twitter-YN (line 365-366).
>
> O.
> RE: GPT Experiments: Can you provide more detailed descriptions of your experiments involving the GPT model?
> - As stated in Section 5.2, we provide the exact version, arguments, and other necessary details in Appendix F. Figures 4-8 detail the exact prompts we use. We are confident stating that  that’s all that is needed to use the GPT API.
>
>     Additionally, we will also release our code (footnote 1).

---

### Official Review · Reviewer_9N5z · 2023-08-13

**Soundness:** 3

**Excitement:**

3: Ambivalent: It has merits (e.g., it reports state-of-the-art results, the idea is nice), but there are key weaknesses (e.g., it describes incremental work), and it can significantly benefit from another round of revision. However, I won't object to accepting it if my co-reviewers champion it.

**Paper Topic And Main Contributions:**

The paper delves into the intriguing challenge of interpreting answers to yes-no questions in the context of social media, specifically Twitter. Given the informal and often ambiguous nature of user-generated content, this is a pertinent issue in the realm of question answering.

**Questions For The Authors:**

- How were the 4,442 yes-no question-answer pairs curated? Were there any specific criteria or filters applied during the collection process?

- Were there any regional or cultural variations in the way users responded to yes-no questions?

**Reasons To Accept:**

- New Corpus Creation: The paper presents a corpus of 4K+ binary question-answer pairs from Twitter. This dataset can serve as a valuable resource for the social media as well as QA research.

- Linguistic Analysis: The papers analyzes the linguistic characteristics of answers, whether they lean towards 'yes', 'no', or are 'unknown'. The linguistic perspective present the nuanced understanding  required for this corpus as well as  the challenges in interpreting user-generated content.

- Detailed Model Evaluation: The paper evaluates the performance of large language models, after fine-tuning and blending with other corpora. The experimentation presents a comprehensive comparison with of the current state-of-the-art models.

**Reasons To Reject:**

Model Details: The paper lacks specifics about model design large language models were tested. A more detailed analysis of the models, visual description of their architectures, and the fine-tuning process would be beneficial for reproducibility and a deeper understanding of the results.

Comparison with Other Platforms: While the focus is on Twitter, it might be interesting to see if the findings are consistent across other social media platforms or if there are platform-specific nuances.

**Reproducibility:**

4: Could mostly reproduce the results, but there may be some variation because of sample variance or minor variations in their interpretation of the protocol or method.

**Reviewer Confidence:**

4: Quite sure. I tried to check the important points carefully. It's unlikely, though conceivable, that I missed something that should affect my ratings.

---

> ### Author Rebuttal · Authors · 2023-08-26
>
> Thank you for your review. We appreciate your comments.
>
> **Reasons to Reject:**
>
> RE: Model Details, visual descriptions of architectures, fine-tuning process, etc.:
> - We cover this content in Section 5, 5.1 and 5.2, including citations and hyperparameter search (train / val / test split, etc.). Appendices D include the specific search space for hyperparameters and the final values. Appendix G includes the detailed prompts (Figures 5-8), GPT versions, and parameters (temperature, frequency penalty, etc.). Regarding reproducibility, please see Tables 9-14 with detailed hyperparameters. Also, note that we will release the dataset and our implementation to replicate our results (footnote 1).
>
>     Based on this comment, we will add more details and move some content from the appendix to the main paper. We appreciate the suggestion.
>
> RE: Comparison with Other Platforms: While the focus is on Twitter, it might be interesting to see if the findings are consistent across other social media platforms or if there are platform-specific nuances.
> - We agree. We believe that it is clear from page 1 that we only work with Twitter and we acknowledge it in the Limitations. Thank you for the suggestion. We will add working with other social media platforms to our research agenda.
>
>
> **Questions for the  authors:**
>
> RE: How were the 4,442 yes-no question-answer pairs curated? Were there any specific criteria or filters applied during the collection process??
> - Section 3.1 explains how the corpus was curated, including our rationale. Section 3.2 includes a shorter version of the annotation guidelines and details the annotation process. Appendix A provides a comprehensive Data Statements that provides more details (annotators, additional rationale, etc.)
>     As explained in those sections, we use filters to identify yes-no questions. As previous work has shown (Section 2), identifying yes-no questions in dialogue is not challenging. The challenging problem is interpreting the answers to those questions.
>
>
> RE: Were there any regional or cultural variations in the way users responded to yes-no questions?
> - Probably yes. This is true of any NLP problem, especially problems that work with dialogues. That said, this question is outside of the scope of the current paper as Twitter does not provide detailed regional or cultural information.

---

### Meta-Review · Area_Chair_ZMdT · 2023-09-14

**Recommendation:** 3

**Metareview:**

This paper introduces a dataset for the task of interpreting responses to yes/no questions asked on Twitter. This can often be challenging because the responses may not explicitly contain the words "yes" or "no", and the interpretation might require commonsense knowledge. The authors created a dataset of 4K+ QA pairs and interpretations representing this task. The reviewers generally agreed that the dataset can serve as a valuable resource. Reviewer 8ozY had concerns about the motivation, experimental methodology, and the organization of the paper, but the authors' responses seem convincing to me.

There are existing datasets for this task, e.g.: BoolQ, but not in this domain (social media, particularly Twitter), making this a potentially useful resource.

---

### Decision · Program_Chairs · 2023-10-07

**Decision:**

Accept-Findings

**Comment:**

This paper introduces a dataset for the task of interpreting responses to yes/no questions asked on Twitter. This can often be challenging because the responses may not explicitly contain the words "yes" or "no", and the interpretation might require commonsense knowledge. The authors created a dataset of 4K+ QA pairs and interpretations representing this task. The reviewers generally agreed that the dataset can serve as a valuable resource. Reviewer 8ozY had concerns about the motivation, experimental methodology, and the organization of the paper, but the authors' responses seem convincing to me.

There are existing datasets for this task, e.g.: BoolQ, but not in this domain (social media, particularly Twitter), making this a potentially useful resource.